# Breaking Left–Right Symmetry by the Interplay of Planar Cell Polarity, Calcium Signaling and Cilia

**DOI:** 10.3390/cells13242116

**Published:** 2024-12-20

**Authors:** De-Li Shi

**Affiliations:** Laboratoire de Biologie du Développement, LBD, CNRS UMR7622, INSERM U1156, Sorbonne Université, F-75005 Paris, France; de-li.shi@upmc.fr

**Keywords:** left–right organizer, left–right asymmetry, cilia, Wnt/PCP, calcium, polycystin, laterality defects

## Abstract

The formation of the embryonic left–right axis is a fundamental process in animals, which subsequently conditions both the shape and the correct positioning of internal organs. During vertebrate early development, a transient structure, known as the left–right organizer, breaks the bilateral symmetry in a manner that is critically dependent on the activity of motile and immotile cilia or asymmetric cell migration. Extensive studies have partially elucidated the molecular pathways that initiate left–right asymmetric patterning and morphogenesis. Wnt/planar cell polarity signaling plays an important role in the biased orientation and rotational motion of motile cilia. The leftward fluid flow generated in the cavity of the left–right organizer is sensed by immotile cilia through complex mechanisms to trigger left-sided calcium signaling and lateralized gene expression pattern. Disrupted asymmetric positioning or impaired structure and function of cilia leads to randomized left–right axis determination, which is closely linked to laterality defects, particularly congenital heart disease. Despite of the formidable progress made in deciphering the critical contribution of cilia to establishing the left–right asymmetry, a strong challenge remains to understand how cilia generate and sense fluid flow to differentially activate gene expression across the left–right axis. This review analyzes mechanisms underlying the asymmetric morphogenesis and function of the left–right organizer in left–right axis formation. It also aims to identify important questions that are open for future investigations.

## 1. Introduction

The formation of embryonic axes is a fundamental process in development. It contributes to shaping the basic body plan of bilaterian organisms. The establishment of the dorso-ventral (D–V) and antero-posterior (A–P) axes is tightly linked, which takes place before and during gastrulation. This is regulated by an interplay of conserved signaling factors that coordinate spatial–temporal gene expression patterns and morphogenetic cell movements [1,2]. The specification of the left–right (L–R) axis occurs at early stages of somite segmentation. The signaling pathways implicated in initiating the L–R pattern are also evolutionarily conserved in metazoa, but differences in the mode of their functions may exist among various species [3,4,5,6]. Although the developing vertebrate embryos seemingly show bilateral symmetry, they already possess L–R asymmetry of gene expression patterns. Importantly, the formation of the L–R axis not only breaks the mediolateral symmetry of the early embryo but also dictates the asymmetric location and morphogenesis of internal organs, which eventually display L–R differences in sizes, shapes, and anatomical dispositions. Therefore, the L–R asymmetry is a common aspect in animals [7,8], particularly in the development of visceral organs, such as the rightward looping of the heart tube and the leftward curvature of the stomach, as well as their precise positioning in the body cavity [9].

How is the L–R asymmetry initiated in the embryo? It is now well established that transient embryonic structures, including the node in mice, the Hensen’s node in chicks, the posterior gastrocoel roof plate in *Xenopus* early neurula, and the Kupffer’s vesicle (KV) in zebrafish, function as L–R organizers and produce instructive cues to break the bilateral symmetry across the mediolateral plane mostly in a cilia-dependent manner [10,11,12,13]. Thus, both motile and immotile cilia within the L–R organizers contribute to initiating the determination of the L–R asymmetry [14]. Among many molecular regulators involved in initiating the L–R asymmetry, the Wnt/planar cell polarity (PCP) pathway and calcium signaling play important conserved roles in the morphogenesis of the L–R organizer and in mediating cilia function to activate a left-sided gene expression program, respectively. Components of the Wnt/PCP pathway coordinate cellular orientations in the L–R organizer with respect to the D–V and A–P axes [9,15,16,17,18]. Subsequently, the clockwise (ventral view) rotational motion of motile cilia within the L–R organizer triggers asymmetric fluid flow and calcium flux on the left side, known as Nodal flow [10,19,20], and signals the determination of the L–R axis [21,22]. As a result, the Nodal–Lefty–Pitx2 gene network is activated on the left side [11]. This differential gene expression establishes the L–R polarity that critically contributes to positioning the asymmetric morphogenesis of organ primordia [23,24,25]. Because the close link between L–R axis determination and asymmetric organogenesis, disrupted formation and function of the L–R organizer can result in laterality defects such as heart malformations [26,27]. While situs solitus refers to the normal anatomy pattern, altered L–R axis development causes situs inversus, heterotaxia, or isomerism, which are congenital disorders with complete reversed, partially inverted, or symmetric positioning of visceral organs, respectively.

This review focuses on two closely linked early events essential for breaking the bilateral symmetry: Wnt/PCP signaling regulates the asymmetric positioning of motile cilia in the L–R organizer, while the leftward Nodal flow sensed by immotile cilia triggers intraciliary calcium transients to initiate L–R axis patterning. It also briefly presents the consequences of mutations affecting genes associated with the L–R organizer on congenital heart malformations. By providing an in-depth analysis of the intricate processes operating in the L–R organizer, this work offers insights into the molecular mechanisms underlying L–R axis development and asymmetric organ morphogenesis.

## 2. The Wnt/PCP Pathway

Wnt signaling is evolutionarily conserved and can be subdivided mainly into canonical (Wnt/ß-catenin) and non-canonical (Wnt/PCP) pathways. Wnt/ß-catenin signaling regulates gene expression and cell fate specification in a manner that is dependent on ß-catenin nuclear translocation. The Wnt/PCP pathway (Figure 1A), however, controls cellular orientation within the plane of an epithelium or a tissue by regulating cytoskeletal rearrangements and/or transcriptional responses [9]. The PCP phenomenon is essentially coordinated by six “core” proteins (Frizzled, Celsr1, Vangl2, Dishevelled or Dvl, Prickle, and Ankrd6), which transduce the signal through downstream effectors including Daam1 (Dishevelled-associated activator of morphogenesis 1), small GTPases of the Rho family, and JNK (Jun N-terminal kinase). In vertebrates, several Wnt ligands (Wnt5 and Wnt11) also contribute to establishing the PCP by binding to membrane receptors (Frizzled) and co-receptors (Ror1/2 or Ryk), although they are not considered as “core” PCP proteins [28]. The “core” PCP proteins form two separate complexes that show characteristic asymmetric distribution on opposite borders of the cell within the tissue plane. Thus, Frizzled, Dvl, and Ankrd6 localize to one side of the cell, while Prickle and Vangl2 are present at the opposite side. Celsr resides on both sides of the cell and functions to propagate polarity information across cells by forming homodimers between adjacent cells (Figure 1B). This asymmetric distribution of “core” PCP protein complexes establishes planar polarization and controls cell polarity during tissue and organ morphogenesis [9,29,30,31].

It has been proposed that there are also other conserved protein complexes acting as important PCP regulators. The heteromeric protocadherins Fat4 and dachsous cadherin-related 1 (Dchs1) are present on opposite cell borders to form a ligand–receptor pair (Figure 1C). This module is considered as the second PCP pathway, at least in *Drosophila*. It is regulated by the Golgi resident transmembrane kinase Four-jointed (Fj), which phosphorylates serine/threonine residues within the extracellular cadherin domains of Fat4 and Dchs1 as they transit through the Golgi [32]. The Scrib (Scrb1 or Scribble1) polarity complex, which consists of Scrib, Discs-large (Dlg), and Lethal-giant larvae (Lgl) proteins, was initially identified as a regulator of apico-basal cell polarity in *Drosophila*, but there is a large body of evidence suggesting that it also acts to coordinate PCP in vertebrates [33]. Therefore, these pathways function cooperatively with or independently of “core” PCP proteins to regulate cell polarity.

## 3. Wnt/PCP Signaling Instructs the Asymmetric Orientation of Motile Cilia in the L–R Organizer

There are two types of ciliated cells in the mouse node [13,20]. Pit cells are located within a depression at the center of the node. Their motile cilia are posteriorly tilted and rotate clockwise to produce leftward fluid flow. Crown cells are located at the periphery of the node and possess immotile cilia. Several components of the Wnt/PCP pathway are critically involved in the asymmetric morphogenesis of the vertebrate L–R organizer, and their restricted distribution makes an important contribution in breaking the bilateral symmetry [11]. In mice, Wnt5a and Wnt5b are expressed in the posterior region of the node; they produce a diffusible gradient which provides instructive signals to polarize node cells along the A–P axis by promoting the anterior localization of three “core” PCP proteins: Vangl1, Vangl2, and Prickle2 [34]. Complementary with this distribution, Dvl2 and Dvl3 are enriched at the posterior side of node cells [11,35]. As a result, this leads to a biased distribution of microtubules and actomyosin networks, thereby restricting posteriorly the positioning of ciliary basal bodies and the tilting of cilia in node cells [36]. Functional studies in different species have firmly established a critical role for PCP proteins in initiating the L–R differences. Knockout of *Dvl2* and *Dvl3* perturbs the posterior positioning of ciliary basal bodies in mouse node cells and impairs the unidirectional fluid flow [35]. Similarly, Vangl1 and Vangl2 are required for the posterior orientation of motile cilia in the mouse node [37,38,39], the zebrafish KV [40], and the *Xenopus* gastrocoel roof plate [37]. Thus, the loss of their function interferes with the left-sided expression of Nodal, Lefty, Sonic hedgehog (Shh), and Pitx2 [37,38,39,41].

Functional interactions between PCP proteins are important for their asymmetric distribution in the L–R organizer. In mice, the anterior localization of Vangl2 is dependent on Prickle1 and Prickle2 [34]. Dachsous protocadherins have been shown to exert a permissive effect on cellular polarization. The combined loss of *Dchs1* and *Dchs2* disrupts the A–P distributions of Vangl1 and Vangl2 in pit cells of the node, inhibiting the left-sided expression of *Nodal* gene in the lateral plate mesoderm without affecting its expression in the node [36]. In *Xenopus*, there is evidence showing that the reciprocal interactions between Prickle3 and Vangl2 are required for their localization to the anterior borders of gastrocoel roof plate cells, which contributes to promoting the growth and posterior tilting of motile cilia [42]. In the zebrafish KV, JNK activity functions in Wnt/PCP signaling to establish the L–R axis. While JNK1 and JNK2 modulate ciliogenesis and cilia length to generate fluid flow in the KV and restrict left-sided expression of the *Nodal*-related gene *southpaw* in the lateral plate mesoderm, JNK3 acts downstream of JNK1 to confine the expression of *pitx2c* on the left side for specification of endodermal organs [43,44]. Overall, these observations demonstrate a critical role for Wnt/PCP signaling in the asymmetric orientation of motile cilia within the L–R organizer. Therefore, the A–P polarization of PCP proteins is translated into L–R asymmetry through biased positioning of motile cilia and their clockwise rotational motion to generate leftward fluid flow [45].

Wnt/PCP signaling also acts in concert with other regulators of the L–R asymmetry to polarize cilia and fluid flow in the L–R organizer. Myo1D is an unconventional myosin that plays an essential role in the control of the L–R asymmetry and the dextral rotation of visceral organs [46,47]. In zebrafish and *Xenopus*, Myo1D and Vangl2 functionally interacts and exert opposing effects to differentially restrict the orientation of cilia within the L–R organizer, thereby shaping a productive fluid flow and promoting the left-sided gene expression in the lateral plate mesoderm [48,49]. However, Myo1D can also function independently of Wnt/PCP signaling. It contributes to the formation of a spherical lumen for correct fluid flow in the zebrafish KV through fluid filling, by promoting directed vacuole trafficking in the apical membrane of KV epithelial cells [50].

## 4. Cilia-Driven Leftward Nodal Flow Initiates the L–R Asymmetry

The A–P polarity of the L–R organizer, created by the differential subcellular localization of PCP proteins and the asymmetric orientation of motile cilia, breaks the bilateral symmetry through cilia-dependent leftward fluid flow and an increase of calcium concentrations [36]. Calcium signaling then triggers differential L–R gene expression patterns and initiates the L–R asymmetry [11]. Subsequently, the left-sided expression of the Nodal–Lefty–Pitx2 network will influence the asymmetric organ morphogenesis [23,24,25]. However, it seems that the vertebrate L–R organizers break the bilateral symmetry in a manner that is both cilia-dependent and independent. While cilia-driven fluid flow clearly contributes to the determination of the L–R asymmetry in *Xenopus*, zebrafish and mice, its function in the chick embryo remains largely elusive (Figure 2).

Nodal flow leads to an increased calcium concentration on the left border of the L–R organizer, and immotile cilia function as flow sensors to initiate asymmetric calcium signaling [51]. Consequently, this results in the left-sided expression of several genes, such as *Lefty* and *Nodal* in mice, *southpaw* in zebrafish, and *Xnr1* in *Xenopus* [20]. In the chick embryo, there is no evidence indicating Nodal flow, suggesting the lack of a requirement for cilia function [52]. However, it has been shown that cilia are present at Hensen’s node between the dorsal epiblast and the ventral endoderm [53]. In addition, increased extracellular calcium levels have been detected on the left side of Hensen’s node, which is correlated with the expression of Nodal and Shh on the left side and fibroblast growth factor 8 (Fgf8) on the right side [54]. Nevertheless, functional assays are necessary to discriminate whether cilia trigger fluid flow and how the flow is sensed to establish asymmetric gene expression around the Hensen’s node. Although the presence, type, and function of cilia in the avian L–R organizer remain controversial, it is relatively well established that asymmetric cell migration plays an important role in breaking the bilateral symmetry. There is evidence that asymmetric bilateral counter-rotating cellular flows, termed as “polonaise” movements, occur prior to the formation of the Hensen’s node and display a right dominance [55]. It has also been shown that cells expressing Shh and Fgf8 around the Hensen’s node undergo rearrangements and exhibit transient leftward movements after the full elongation and before the regression of the primitive streak, leading to asymmetric expression of these genes with respect to the midline [56,57]. Interestingly, an appropriate level of N-cadherin is important for the leftward movements to occur within this specific time window [58].

The importance of Nodal flow in breaking the bilateral symmetry has been well documented in different species [59,60]. Pioneer works show that disruption of Nodal flow in mice due to the absence of motile cilia causes randomization in the expression of flow target genes and prevents the establishment of the L–R asymmetry [21]. Interestingly, a sufficiently rapid artificial rightward flow can counteract the intrinsic leftward Nodal flow and cause situs inversus in wild-type embryos, whereas an artificial leftward flow can direct situs solitus in mutant mice with defective motile cilia [22]. Surprisingly, although 200–300 rotational cilia are present in the mouse node, there is evidence showing that as few as two motile cilia are sufficient to generate a local leftward flow and promote the formation of the L–R asymmetry [61]. This seems to suggest that there is a threshold of a weak, but nevertheless sensitive, unidirectional Nodal flow that is sufficient to trigger the left signal. Analyses in zebrafish indicate that 30 out of 200 functional motile cilia are required for situs solitus [62]. Since fewer motile cilia would only generate a weak flow, the above observations raise the possibility that Nodal flow may be perceived as a mechanical force. Consistently, recent studies suggest that the mechanical properties of Nodal flow are responsible for breaking the bilateral symmetry. In the zebrafish KV, fluid replacement experiments indicate that the L–R organizer is sensitive to fluid dynamics instead of fluid content [63]. However, as discussed below, there are also arguments supporting the existence of a morphogen gradient generated by fluid flow across the mediolateral plane. Indeed, although the observation that two rotating cilia are able to trigger fluid flow may favor the mechanical force model, it cannot exclude the possibility that there may be still left-sided transport of chemical cues.

Intriguingly, the clockwise rotational movement of motile cilia in the mouse node not only depends on their posterior tilting coordinated by Wnt/PCP signaling but is also regulated by the absence of the radial spoke, a multi-unit protein structure present at the center of the axoneme. The knockout of *Rsph4a*, which encodes a component of the axonemal radial spoke head, transforms the planar beating of airway cilia into a clockwise rotation and promotes the clockwise rotation of node cilia [64,65]. This observation leads to the speculation that mouse node cilia have lost radial spokes during evolution and lack the central structure, although fewer 9 + 2 type cilia are still present [64,65]. There are also several lines of evidence indicating that the disruption of structural and functional components of axonemal microtubules causes defective unidirectional rotation of cilia. Disturbing the regular arrangement of doublet microtubules in mouse node cilia prevents their stable clockwise rotation [64]. The mutations of different axonemal dynein proteins lead to a loss of cilia motility and L–R patterning defects in mice and humans [66,67,68]. Similarly, kinesin family members of microtubule motors are required for ciliary morphogenesis. Thus, the knockout of *Kif3A* or *Kif3B* in mice leads to the absence of all cilia in the node and randomized L–R asymmetry [21,69,70]. The defective cilia phenotype can be observed at E7.5, before the earliest expression of L–R asymmetric genes [69], suggesting that motile and immotile cilia contribute to the initial determination of the L–R asymmetry. In addition, overexpression of a mutated tubulin protein in *Xenopus* perturbs the sidedness of L–R asymmetric gene expression, but whether this affects cilia formation and motility is not clear [71].

## 5. Calcium Signaling in L–R Axis Patterning

The left-sided activation of calcium signaling plays a conserved role in L–R asymmetry formation, but the molecular mechanisms are still under intensive investigations. In the mouse node, calcium signals can be detected on the left margin of the node coincident with Nodal flow [51,72,73]. Pharmacological inhibition of calcium signals alters early asymmetric gene expression and disrupts L–R axis formation [72]. In the zebrafish KV, left-biased intraciliary calcium oscillations are linked to the activation of downstream L–R signaling and molecular asymmetry [74]. Reducing calcium influx or the suppression of intraciliary calcium oscillations disrupts L–R patterning [74,75]. Functional analyses of calcium-sensitive channels also strongly implicate calcium signaling in L–R axis development. The calcium-permeable cation channel Polycystin-2 (PC2), encoded by the *Pkd2* gene, is a six-pass transmembrane protein and a member of the transient receptor potential (TRP) channel family. Mutations of the *PKD2* gene in humans are responsible for autosomal dominant polycystic kidney disease (ADPKD). PC2 not only functions as a cation-permeant and calcium-sensitive channel, but also acts as a regulator of other channels, contributing to intracellular signaling [76]. It is present in both motile and immotile cilia but seems to be preferentially localized to immotile cilia in crown cells [51]. PC2 is clearly involved in modulating calcium signaling to differentially regulate L–R gene expression (Figure 3). In zebrafish and mice, the loss of PC2 function disrupts asymmetric calcium signaling associated with the defective expression of L–R genes [51,77,78,79]. The knockdown of *Pkd2* in *Xenopus* also causes the absence of fluid flow and prevents the left-sided activation of a Nodal signaling cascade [80]. Nevertheless, it seems that PC2 exerts different effects on Nodal expression in mice and zebrafish. Mouse *Pkd2* mutants lack a left-specific calcium expression at the node and show a complete absence of Nodal expression in the lateral plate mesoderm [51,77], while zebrafish *pkd2* mutants display a bilateral activation of *southpaw* expression in the lateral plate mesoderm [78]. Thus, PC2 in mice triggers an asymmetric calcium transient to activate the left-sided Nodal expression, while it may function to restrict the Nodal expression on the left side in zebrafish. In addition, it has also been suggested that *pkd2* deficiency in zebrafish may reduce cilia length and indirectly affect Nodal flow dynamics [81].

Although the elevated calcium signaling on the left side of vertebrate L–R organizers is indispensable for the differential expression of laterality genes across the L–R axis, the mechanism by which cilia perceive this information is not fully understood and remains a subject of debate. Using fluorescent calcium indicators, asymmetric intraciliary calcium oscillations have been observed in the zebrafish KV and the mouse node, which are tightly linked to L–R asymmetric gene expression and determination [72,73,74]. Despite the presence of calcium transients on both sides of the mouse node, quantitative analysis indicates that the mean frequency of calcium spikes is two-fold higher on the left side than on the right side [73]. Both in zebrafish and mice, the preferential left-sided calcium oscillations are clearly dependent on the function of PC2 [73,74]. It is well established that calcium oscillations control the efficiency and specificity of gene expression in a variety of processes [82]. Several mechanisms may be involved in differentially regulating gene expression on the left and right sides. Signaling pathways regulated by Ca^2+^/calmodulin-dependent protein kinase II (CaMKII), which is localized to cilia and required for L–R in zebrafish [83], may be activated by intraciliary calcium oscillations to mediate gene transcription. Calcium ion is also an important regulator of protein charge and conformation [84]; thus, it can function as a potent second message to regulate many signaling pathways, leading to the transcriptional, post-transcriptional, or post-translational activation of gene expression. Other targets of the intraciliary calcium signaling may include adenylyl cyclase (AC) which influences the level of cAMP [85], as well as Inversion (Inv), which contains Calmodulin-binding IQ domains and is involved in the control of L–R asymmetry [86,87]. There is also evidence that the intracellular tail of PC1 becomes cleaved following mechanical stimuli; PC2 modulates its nuclear translocation and transcriptional activity [88,89]. Despite this progress, future studies are necessary to identify the targets of these signaling mechanisms and determine how they differentially regulate gene expression for L–R axis specification. It should be mentioned that there is still some discrepancy regarding calcium increase in primary cilia. One study has failed to detect calcium transients in the mouse node, raising the possibility that ciliary function may be regulated by calcium propagation from the cytoplasm into the cilium [90].

## 6. Mechanosensor and Chemosensor of Fluid Flow

How is Nodal flow sensed in the L–R organizer? At present, two hypotheses have been proposed to explain this important phenomenon: mechanosensor and chemosensor models [20]. The mechanosensor model suggests that bending of the axoneme in immotile cilia caused by fluid flow can trigger the activation of intraciliary calcium flux and subsequent asymmetric gene expression on the left side. Indeed, mechanical oscillation of cilia stimulated by optical tweezers is sufficient to induce intraciliary calcium transients in a PC2-dependent manner [91,92]. In the mouse node, it seems that PC2 is enriched at the dorsal side of immotile cilia [93]. Fluid flow causes the asymmetric deformation of immotile cilia along the D–V axis, and calcium flux can be induced by ventrally directed mechanical force [92]. In the zebrafish KV, PC2 also mediates intraciliary calcium transients, and interestingly, situs inversus in mutant fish lacking motile cilia can be rescued by applying mechanical force to immotile cilia [91]. Polycystin-1 (PC1), an eleven-pass transmembrane protein, is also a member of the TRP channel family and is responsible for ADPKD in humans [94]. It may associate with PC2 in a heteromeric complex to mediate ion permeation of the cilia, although PC2 can also function as a homomeric complex [20]. The PC1 class of the TRP channel includes PC1L1, PC1L2, and PC1L3, encoded by *Pkd1l1*, *Pkd1l2*, and *Pkd1l3*, respectively. The knockout of *pkd1l1* in zebrafish leads to the bilateral activation of Nodal expression, suggesting that PC1L1 normally functions to restrict left-sided Nodal expression [95]. However, the loss of PC1L1 and PC2 in mice produces opposite effects. Mechanistically, PC1L1 may antagonize the function of PC2 and mediate a response to Nodal flow through mechanosensation [95]. These findings demonstrate a mechanosensory function of immotile cilia, which interprets Nodal flow in a manner that requires the correct localization of PC2. Nevertheless, as aforementioned, experiments targeting calcium sensors to the cytoplasm and cilia suggest that mechanical forces do not directly evoke intraciliary calcium signaling, thus questioning whether primary cilia in general are calcium-responsive sensors [90]. Therefore, further investigations are necessary to determine the regulatory mechanisms of ciliary mechanosensation.

The chemosensor model proposes that Nodal flow generates a gradient of putative morphogens or activators of laterality genes to specify the initial L–R axis [96]. It has been shown that Shh- and retinoic acid-containing secreted extracellular vesicles, termed as “Nodal vesicular parcels”, are transported to the left side of the mouse node by fluid flow, thus establishing a L–R gradient of morphogens [97]. Intriguingly, these extracellular vesicles also contain PC1L1 protein, which is transferred to crown cells on the left side of the node where it forms a functional complex with PC2 to mediate calcium elevation on the left margin [98]. This is consistent with several lines of evidence showing that PC1L1 and PC2 establish the L–R asymmetry through physical interaction and interdependent colocalization to the cilium [99,100]. Although it is unclear how PC1L1-containing exosomes are secreted into the node cavity, the PC1L1-PC2 complex clearly plays a chemosensory role to mediate calcium elevation and signaling [20]. Thus, how cilia sense Nodal flow to activate left-sided gene expression remains a topic of heated debate. There may also be a species-specific reception and interpretation of Nodal flow. In zebrafish, it has been documented that the very limited numbers of immotile cilia present in the KV are not sufficient to mediate mechanosensing for robust determination of the L–R asymmetry; however, motile cilia may be able to mechanically sense their own motion to generate an asymmetric response or absorb secreted signaling molecules first present on the left side [101]. Based on experimental data and computational modeling [101], it cannot be excluded that chemical signal and mechanical stimulation may function in parallel to establish the L–R asymmetry.

## 7. Flow-Induced Differential L–R Gene Expression

Although there are many unanswered questions in both mechanosensor and chemosensor models [20,102], Polycystin proteins are clearly essential in sensitizing crown cells and evoking calcium signaling. The link between calcium signaling and asymmetric gene expression is also enigmatic, but early molecular events triggered by flow-mediated signal(s) are beginning to be elucidated. Dand5, also known as Cerl2 or Cer2, functions as an extracellular antagonist of Nodal protein by preventing the formation of heterodimers between Nodal and Gdf1 [103]. It is the first asymmetrically expressed gene involved in L–R patterning [61,104], and its mRNA is selectively degraded in crown cells on the left side of the mouse node, resulting in expression on the right side [105]. Interestingly, the RNA-binding protein, Bicc1 (Bicaudal C), which is also required for L–R patterning [106], binds to the 3′-UTR of *Dand5* mRNA to promote its degradation at the left side in two-somite stage embryos [107,108]. As a result, this leads to an increased Nodal activity and the consequent expression of Nodal, Lefty, and Pitx2 on the left side of the lateral plate mesoderm [11]. Thus, Dand5 represents an early flow target gene with an important role in establishing the gene expression network for L–R patterning [109,110]. In *Bicc1* mutant mice, the expression of *Lefty1*, *Lefty2*, and *Pitx2* becomes bilateral, inverted, or absent [106]. Similarly, the loss of Dand5 in mice causes the bilateral and right-sided expression of *Lefty2* and *Pitx2* [105]. Although the midline expression of *Lefty1* does not seem to be affected, it becomes more pronounced in the node and predominantly localizes to the posterior side [87,105,106]. Since Lefty1 inhibits Nodal function, this ectopic expression of Lefty1 in the node should perturb the onset of asymmetric Nodal signaling in the lateral plate mesoderm [87]. Therefore, the post-transcriptionally controlled local degradation of *Dand5* is critical for the proper activation of the Nodal program in the lateral plate mesoderm [111]. Nevertheless, the mechanisms by which ciliary signaling activates or represses left-sided gene expression require further investigation.

There is also evidence that Wnt/ß-catenin signaling contributes to regulating the asymmetric expression of Dand5 [112] because Wnt-Dand5 interlinked feedback loops can enhance the leftward flow [113]. Wnt3 exhibits L–R differences in expression and promotes the decay of *Dand5* mRNA in crown cells on the left side, while Dand5 can also induce Wnt3 degradation [113]. Thus, the differential activation and repression of gene expression establishes L–R embryonic polarity that will influence asymmetric organ morphogenesis.

## 8. Randomization of L–R Asymmetry and Laterality Defects

The molecular asymmetry established in the early embryos contributes in part to the asymmetric organ development. Therefore, the mutation or dysregulation of genes involved in the morphogenesis of the L–R organizer is closely linked to laterality disorders [26,27]. Because the heart is the first formed functional organ and undergoes asymmetric morphogenesis during development, it is not surprising that defective function of the L–R organizer frequently causes heterotaxy and congenital heart disease [114]. As aforementioned, Wnt/PCP signaling is required for the asymmetric positioning of cilia in the L–R organizer. Its dysfunction is tightly associated with congenital disorders affecting proper organogenesis [9,115]. Randomized positioning of cilia in the node due to mutations of PCP genes in mice, such as *Vangl2*, impairs Nodal flow, leads to reversal of *Lefty1/2* expression and bilateral *Pitx2* expression in the lateral plate mesoderm, and causes defective rightward looping of the heart tube [38,39]. The Zic3 transcription factor is expressed in the mouse node but not in the heart; however, its loss of function not only affects L–R patterning but also leads to heart laterality disorders [116]. There is evidence that Zic3 functions early in L–R organizer development by regulating the expression of PCP genes [117,118]. In humans, frameshift, missense, and nonsense mutations of the *ZIC3* gene cause X-linked situs abnormalities ranging from heterotaxy to situs ambiguus or situs inversus [119]. Myo1D regulates cilia orientation and Nodal flow in concert with or independently of Wnt/PCP signaling [47,48,49]. A rare novel missense variant of *MYO1D* gene has been linked to polysplenia syndrome presenting visceral heterotaxy and left isomerism [120]. In addition, single cell sequencing of the mouse L–R organizer has identified potential novel heterotaxy-related genes, which may be useful for studying the genetic cause of laterality disorders [121]. Thus, functional analysis of these genes not only helps decipher the mechanism underlying laterality establishment but also contributes to understanding the genetic cause of heterotaxy.

The loss of structural or signaling components in L–R organizer-related cilia also severely affects asymmetric cardiac morphogenesis. Indeed, mutagenesis screen in mice indicate that congenital heart disease (CHD) genes are related to cilia function or cilia-mediated signal transduction, such as Shh, Wnt/PCP components, calcium signaling, and vesicular trafficking. Importantly, many of these genes overlap with de novo coding mutations identified in human CHD patients [122,123]. For example, homozygous splicing and missense mutations of *PKD1L1* in humans are associated with heterotaxy or situs inversus totalis and congenital cardiac malformations [124]. It is also worth mentioning that mutations of flow target genes, such as *PITX2*, may cause CHD in the absence of laterality defects, suggesting an important and broad role of L–R patterning in the pathogenesis of CHD [122,123]. In addition, dynein dysfunction, as a cause of primary ciliary dyskinesia and other ciliopathies, is clearly associated with laterality defects in humans [125]. Although variants with a loss of kinesin gene function are responsible for many so-called ‘kinesinopathies’ presenting with congenital malformations [126], further investigations are necessary to determine whether they are linked to situs abnormalities.

## 9. Conclusions and Perspectives

Extensive studies have significantly contributed to understanding the intricate processes that establish the L–R asymmetry, although there are still many intriguing unanswered questions (Figure 4). Importantly, the L–R organizer is indispensable for breaking the initial bilateral symmetry. It is well established that the conserved Wnt/PCP signaling pathway contributes to setting up the biased orientation and rotational motion of motile cilia within the L–R organizer in most species. However, whether there exists an interaction between Wnt/PCP signaling and cilia in initiating the bilateral asymmetry in the chick embryo remains largely elusive and requires further functional investigations. Another key issue concerns the regulatory mechanisms by which asymmetric calcium signaling is initiated within the L–R organizer. Although the two-cilia model suggests that immotile cilia sense Nodal flow generated by motile cilia [127], the identity of the flow input remains a subject of hot debate. In the mouse node, for example, several studies suggest the existence of left-sided intraciliary calcium oscillations triggered by Nodal flow, which are closely linked to the determination of the L–R asymmetry [72,73,74]; however, there are also experiments indicating the absence of calcium-responsive sensors in immotile cilia [90]. Adding to this complexity, it remains to be determined whether Nodal flow is sensed as mechanical forces or chemical cues. The fact that intraciliary calcium flux can be initiated by direct application of mechanical stimulus to a cilium, and that the weak unidirectional flow triggered by two rotating cilia is sufficient to break L–R symmetry, seems to support the mechanosensory model [61,91,92]. On the other hand, it should take into account the possibility that morphogen-based chemical cues may function independently of and/or in parallel with mechanical forces to initiate left–right asymmetric gene expression [95,96]. Therefore, the regulatory mechanisms of ciliary signaling await further investigations. There exists also a gap in understanding the molecular pathways connecting elevated calcium levels and downstream gene expression events. PC1L1 and PC2 may be important for mediating calcium signaling in regulating the expression of flow target genes, although the mechanism remains unclear. It is known that Nodal flow leads to the leftward degradation of *Dand5* mRNA, thereby preventing the expression of Dand5 protein and its antagonizing activity on Nodal signaling on the left side [113]. The post-transcriptional regulation mediated by RNA-binding proteins may contribute to the differential expression of Dand5 [106,108]. For a better mechanistic understanding of how calcium signaling is initiated and how it differentially activates or represses gene expression on the left or right side, it is important to develop new tools for monitoring intraciliary calcium changes that may be triggered by flow-associated mechanical and/or chemical cues. Further studies are also required to identify intraciliary calcium signaling networks responsible for L–R asymmetric gene expression at transcriptional, post-transcriptional, and post-translational levels.

The mutations of genes involved in L–R axis formation are closely associated with laterality defects in animal models and in humans. However, besides microtubule motors dyneins, mutations of only a few genes functioning in the L–R organizer have been associated with human laterality defects. The identification of novel genes expressed in this transient embryonic structure could facilitate genetic screening of congenital malformations in humans. A better understanding of the genetic cascade that controls L–R axis formation not only will contribute to deciphering the mechanism underlying asymmetric organ morphogenesis, but should also help to develop strategies for the diagnosis of congenital disease, particularly heart malformations.

## Figures and Tables

**Figure 1 cells-13-02116-f001:**
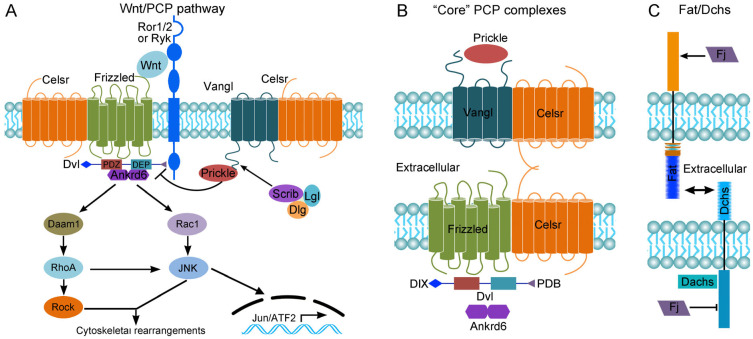
The Wnt/PCP pathway. (**A**) Non-canonical Wnt ligands bind to receptor and co-receptor complexes formed by Frizzled with Ror1/2 or with Ryk to activate downstream effectors via Dvl, regulating cytoskeletal rearrangements or transcriptional responses. The Scrib/Dlg/Lgl complex can regulate Vangl asymmetric localization. (**B**) Cell polarity is established by the interaction between two “core” PCP protein complexes that are distributed on opposite cell borders. Vangl and Prickle localize to the anterior side, while Frizzled, Dvl, and Ankrd6 are distributed at the posterior side. Celsr is present on both sides of the cell to propagate polarity information. DIX, DIX domain; PDB, PDZ domain-binding. (**C**) The Fat/Dchs polarity module. Fat and Dchs protocadherins form heterodimers between adjacent cells and function as a ligand–receptor pair. Phosphorylation of Fat by the Four-jointed (Fj) kinase enhances its binding with Dchs, but phosphorylation of Dchs by Fj reduces its interaction with Fat. Dachs is an unconventional myosin that interacts with Dchs and functions as a key effector of Fat/Dchs signaling.

**Figure 2 cells-13-02116-f002:**
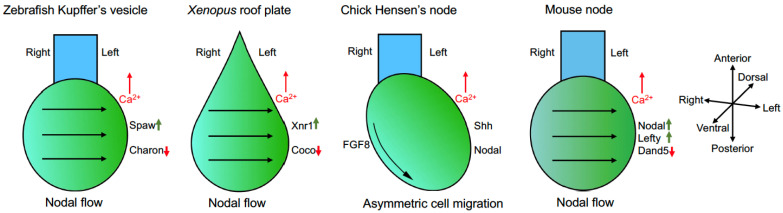
The L–R organizers in vertebrates. The zebrafish Kupffer’s vesicle, *Xenopus* posterior roof plate, chick Hensen’s node and mouse node are L–R organizers that function to establish the L–R asymmetry. By cilia-driven Nodal flow that triggers an increased calcium concentration on the left side (zebrafish, *Xenopus* and mouse) or by asymmetric cell migration (chick), the organizers activate differential gene expression across the L–R axis. As an outcome of calcium signaling, the reduced activity of Nodal signaling antagonists (Charon, Coco, and Dand5) leads to an increased expression of Nodal pathway genes on the left side of the lateral plate mesoderm. Upward pointing arrows indicate an increase of concentration or expression; downward arrows indicate a decreased expression.

**Figure 3 cells-13-02116-f003:**
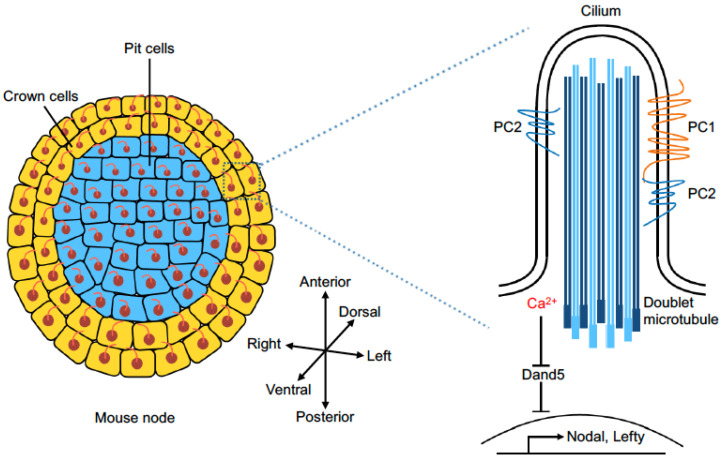
Organization and signaling of the mouse L–R organizer. Located at the posterior end of the notochord, the mouse node consists of pit cells (blue) at the central depression and crown cells (orange) at the periphery. The clockwise (ventral view) rotational movements of posteriorly tilted motile cilia on pit cells generates Nodal flow and causes an increased calcium concentration on the left side. Primary cilia on crown cells sense Nodal flow to activate calcium signaling, which represses the left-sided expression of Dand5 through a post-transcriptional mechanism. Polycystin 1 (PC1) and Polycystin 2 (PC2) are enriched in immotile cilia. They may form a heteromeric ion channel complex to mediate flow-induced activation or repression of gene expression by functioning as a mechanosensor and/or chemosensor.

**Figure 4 cells-13-02116-f004:**
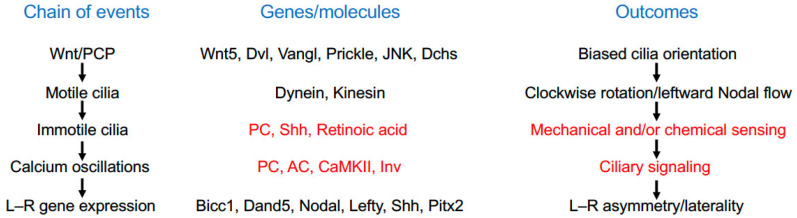
Brief summary of the temporal process leading to the determination of the L–R asymmetry. Further studies for elucidating the underlying mechanisms of ciliary functions are highlighted in red. The list of genes involved in the chain of events are not exhaustive. PC, Polycystin; AC, adenylyl cyclase; CaMKII, Ca^2+^/calmodulin-dependent protein kinase II; Inv, Inversion.

## Data Availability

No new data were created in this work.

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
