# Peer review of "Breaking Left–Right Symmetry by the Interplay of Planar Cell Polarity, Calcium Signaling and Cilia"

_cells, 2024, doi:10.3390/cells13242116_

Round 1
Reviewer 1 Report
Comments and Suggestions for Authors
The author discussed old and new topics related to Left-Right Asymmetry. Recently, a review addressing a similar topic was published (doi: 10.2142/biophysico.bppb-v21.0018). However, this manuscript remains important because it covers a broader range of topics.
To publish this manuscript, a few revisions are necessary. First, the manuscript seems to slightly overlook the role of Lefty. Why did the author not include Lefty in the list of genes that interfere with left-sided expression due to the loss of Vang-like function (Ref. 37)? Additionally, the scheme of Lefty gene expression should be explained in more detail in Chapter 7.
Second, in Figure 3, the author described signaling in "the mouse," but the figure includes genes from other species (e.g., Spaw from fish, Xnr1 from frogs, Charon from fish, and Coco from frogs). The figure should be revised to include only mouse genes.
Author Response
The author discussed old and new topics related to Left-Right Asymmetry. Recently, a review addressing a similar topic was published (doi: 10.2142/biophysico.bppb-v21.0018). However, this manuscript remains important because it covers a broader range of topics.
Response:
Thank you for the positive assessment of this work and for suggestions to improve the manuscript. All issues have been addressed in the revised manuscript. The recent review (doi: 10.2142/biophysico.bppb-v21.0018) highlights the function of cilia in mechanical regulation during left-right determination. It is relevant to the present work and is discussed in the revised manuscript.
To publish this manuscript, a few revisions are necessary. First, the manuscript seems to slightly overlook the role of Lefty. Why did the author not include Lefty in the list of genes that interfere with left-sided expression due to the loss of Vang-like function (Ref. 37)? Additionally, the scheme of Lefty gene expression should be explained in more detail in Chapter 7.
Response:
Lefty is included in the list of genes with disrupted left-sided expression following loss Vangl1 and Vangl2.
The expression of Lefty genes following loss of Bicc1 and Dand5 is explained in more detail in the revised manuscript (lines 395-405).
Second, in Figure 3, the author described signaling in "the mouse," but the figure includes genes from other species (e.g., Spaw from fish, Xnr1 from frogs, Charon from fish, and Coco from frogs). The figure should be revised to include only mouse genes.
Response:
Figure 3 now only includes mouse genes. The chain of events downstream of calcium is also slightly modified to reflect more accurately the discussion in the text.
Reviewer 2 Report
Comments and Suggestions for Authors
The author addresses an important topic in need of an analytical, synthetic review. Overall, the review is clearly written and informative. However, the major critique is that it is not written as a critical, analytical review, but rather a more factual, encyclopedic document. There are plenty of these documents already available; the purpose of a review is to provide a more analytical approach to the topic.
Fig. 1: the figure is too simplistic. The figure should contain the other components discussed in line 106 onward. In addition, the diagram should illustrate a stereotypical cell and delineate the distribution of the anterior and posterior axes to illustrate the asymmetric distribution.
There is very little in this review that discusses the mechanisms by which the calcium signal is transduced, a very important and cutting-edge topic. What is the frequency and amplitude of the calcium signal? What exactly constitutes an "elevated" signal? Are there thresholds? Exactly how is it transduced? What downstream genes are activated, transcriptionally or translationally? How exactly does Ca activate the downstream genes indicated in Figure 3? How exactly is the calcium signal sensed/transduced to lead to this new transcription? Far more detail and analysis is required to make this a robust review.
Subsection 6 is a section of the paper that is written in a more similar fashion to a review. This section is well done because it reviews the literature and delineates where there is agreement and no agreement. The entire paper should be written in this style. It is the job of the author to review the literature; to be very specific (naming groups and authors) about: the important topics in the field and where there is agreement (and the supporting evidence); where there is disagreement (and why, the evidence leading to this), and perhaps most importantly, areas where there is a lack of research.
The "Conclusions and Perspectives" section of the paper needs to be expanded. This is the major job of review papers, namely to provide a critical assessment of the field and future directions, with some specificity. The author should provide much more and specific detail on future avenues of research. In the conclusion, there should also be a summary of areas of agreement among investigators, disagreement among current research, and areas where there is a lack of research. The author should delineate what should be done! This is the job of a review.
The paper would also benefit from a figure that covered the entire temporal process (with question marks delineating areas of disagreement or insufficient research).
Minor points:
l 34-35: awkward phrasing: rephrase "more late stages" [this phrase occurs a few time throughout the paper].
Author Response
Fig. 1: the figure is too simplistic. The figure should contain the other components discussed in line 106 onward. In addition, the diagram should illustrate a stereotypical cell and delineate the distribution of the anterior and posterior axes to illustrate the asymmetric distribution.
Response:
Thank you for the suggestion. This figure is modified to illustrate the asymmetric localization “core” PCP complexes and include other polarity modules (Scrib/Dlg/Lgl and Fat/Dchs), along with a more detailed legend.
There is very little in this review that discusses the mechanisms by which the calcium signal is transduced, a very important and cutting-edge topic. What is the frequency and amplitude of the calcium signal? What exactly constitutes an "elevated" signal? Are there thresholds? Exactly how is it transduced? What downstream genes are activated, transcriptionally or translationally? How exactly does Ca activate the downstream genes indicated in Figure 3? How exactly is the calcium signal sensed/transduced to lead to this new transcription? Far more detail and analysis is required to make this a robust review.
Response:
Thank you for raising these insightful comments, which help to improve the review by providing a more detailed discussion on calcium signal in regulating asymmetric gene expression. Currently, many of these important questions remains unanswered. However, I have made efforts to synthesize relevant works in the literature. In particular, a new paragraph (lines 290-318) that discusses some of the above questions, including left-right differences in calcium oscillations, possible calcium signaling pathways, and discrepancy or future investigations in ciliary signaling, is included in section 5 of the revised manuscript.
Subsection 6 is a section of the paper that is written in a more similar fashion to a review. This section is well done because it reviews the literature and delineates where there is agreement and no agreement. The entire paper should be written in this style. It is the job of the author to review the literature; to be very specific (naming groups and authors) about: the important topics in the field and where there is agreement (and the supporting evidence); where there is disagreement (and why, the evidence leading to this), and perhaps most importantly, areas where there is a lack of research.
Response:
Thank you for the positive assessment of this section and for recommendations to improve the review. Modifications in other sections have been made to discuss important areas where there is disagreement and gap in the research, by citing the specific works. Examples of these modifications concern in particular the conservation of cilia function in chick (lines 193-195, 204-207) and debate related to intraciliary oscillations (lines 312-318, 354-358).
The "Conclusions and Perspectives" section of the paper needs to be expanded. This is the major job of review papers, namely to provide a critical assessment of the field and future directions, with some specificity. The author should provide much more and specific detail on future avenues of research. In the conclusion, there should also be a summary of areas of agreement among investigators, disagreement among current research, and areas where there is a lack of research. The author should delineate what should be done! This is the job of a review.
Response:
This section is expanded by further discussing discrepancies and gaps in the research field with more detail. For example, future studies are required to identify intraciliary calcium signaling networks and develop new tools to monitor intraciliary calcium changes triggered by mechanical and/or chemical cues (lines 492-498).
The paper would also benefit from a figure that covered the entire temporal process (with question marks delineating areas of disagreement or insufficient research).
Response:
Thanks for the helpful suggestion. A new figure (Figure 4) is included in the revised manuscript with important topics requiring further investigations indicated.
l 34-35: awkward phrasing: rephrase "more late stages" [this phrase occurs a few time throughout the paper].
Response:
They are modified (lines 34-35 and line 158).
Reviewer 3 Report
Comments and Suggestions for Authors
The review focuses on the analysis of the fundamental process of the formation of the embryonic left-right axis. The currently accepted hypothesis describing this process is based on the two-cilia model, which suggests that immotile cilia sense leftward Nodal fluid flow generated in the cavity of the left-right organizer by the rotational motion of motile cilia and thus initiates the left-right asymmetry. On the whole, this work offers insights into the molecular mechanisms underlying left-right axis development. Author examine in detail the signaling pathways that subsequently lead to the asymmetric organ morphogenesis and discuss how fluid flow could be sensed — by mechanosensors or chemosensors. However, it seems to me that the text of the article would be significantly improved if it also paid attention to one more important aspect, namely the mechanisms underlying the root cause of all this chain of events by providing the synchronous motility of the cilia where it is needed. Indeed, the manuscript mentions that randomized positioning of cilia in the node due to certain mutations impairs Nodal flow and causes defective rightward looping of the heart tube. It also mentions that disruption of this flow due to the absence of motile cilia prevents the establishment of left-right asymmetry, and that although 200–300 rotational cilia are present in the mouse node, there is evidence showing that as few as just two motile cilia are sufficient to generate a local leftward flow and promote the formation of left-right asymmetry. It follows that the key event, without which all the described signaling pathways are meaningless, is the synchronous movement of the cilia on pit cells. The structure of cilia, i.e., the axoneme of 9+2 microtubules, as well as the basic scheme of its motility, i.e., the sliding of axonemal microtubules by ciliary (axonemal) dyneins powered by ATP hydrolysis, are well known. If there is any information about mutations of microtubule motors or other proteins that lead to immotile cilia on pit cells or ciliopathies, and about the influence of this on the left-right asymmetry and embryonic development, this should be added and discussed.
Author Response
The review focuses on the analysis of the fundamental process of the formation of the embryonic left-right axis. The currently accepted hypothesis describing this process is based on the two-cilia model, which suggests that immotile cilia sense leftward Nodal fluid flow generated in the cavity of the left-right organizer by the rotational motion of motile cilia and thus initiates the left-right asymmetry. On the whole, this work offers insights into the molecular mechanisms underlying left-right axis development. Author examine in detail the signaling pathways that subsequently lead to the asymmetric organ morphogenesis and discuss how fluid flow could be sensed — by mechanosensors or chemosensors. However, it seems to me that the text of the article would be significantly improved if it also paid attention to one more important aspect, namely the mechanisms underlying the root cause of all this chain of events by providing the synchronous motility of the cilia where it is needed. Indeed, the manuscript mentions that randomized positioning of cilia in the node due to certain mutations impairs Nodal flow and causes defective rightward looping of the heart tube. It also mentions that disruption of this flow due to the absence of motile cilia prevents the establishment of left-right asymmetry, and that although 200–300 rotational cilia are present in the mouse node, there is evidence showing that as few as just two motile cilia are sufficient to generate a local leftward flow and promote the formation of left-right asymmetry. It follows that the key event, without which all the described signaling pathways are meaningless, is the synchronous movement of the cilia on pit cells. The structure of cilia, i.e., the axoneme of 9+2 microtubules, as well as the basic scheme of its motility, i.e., the sliding of axonemal microtubules by ciliary (axonemal) dyneins powered by ATP hydrolysis, are well known. If there is any information about mutations of microtubule motors or other proteins that lead to immotile cilia on pit cells or ciliopathies, and about the influence of this on the left-right asymmetry and embryonic development, this should be added and discussed.
Response:
Thank you for the positive assessment of this work and for insightful questions to help improve the manuscript. To address the above questions, a new paragraph discussing the clockwise rotational movement of motile cilia as well as structural and functional components of cilia (such as tubulin, dynein and kinesin) involved in left-right determination is included in section 4 (lines 240-258). In addition, the association of dynein and kinesin mutations with laterality defects in humans is discussed in section 8 (lines 447-451). The revised manuscript also includes a new figure (Figure 4) illustrating the chain of events initiated by motile cilia with genes/molecules involved, such as dynein, kinesin and polycystin (see Figure 4). These modifications should help further understand the molecular and structural bases of cilia function in breaking the left-right symmetry.